# Structure–Activity Relationship of PAD4 Inhibitors and Their Role in Tumor Immunotherapy

**DOI:** 10.3390/pharmaceutics16030335

**Published:** 2024-02-28

**Authors:** Yijiang Jia, Renbo Jia, Ayijiang Taledaohan, Yanming Wang, Yuji Wang

**Affiliations:** 1Department of Medicinal Chemistry, School of Pharmaceutical Sciences of Capital Medical University, 10 Xi Tou Tiao, You An Men, Beijing 100069, China; jiayijiang@ccmu.edu.cn (Y.J.); jiarenbo@ccmu.edu.cn (R.J.); ayijiang227@ccmu.edu.cn (A.T.); 2Department of Medicinal Chemistry, Beijing Area Major Laboratory of Peptide and Small Molecular Drugs, Beijing Laboratory of Biomedical Materials, Engineering Research Center of Endogenous Prophylactic of Ministry of Education of China, 10 Xi Tou Tiao, You An Men, Beijing 100069, China; 3School of Life Sciences, Henan University, Kaifeng 475004, China

**Keywords:** review, PAD4, PAD4 inhibitor, structure–activity relationship, antitumor immunity, combination therapy strategies, drug resistance

## Abstract

Protein arginine deiminase 4 (PAD4) plays an important role in cancer progression by participating in gene regulation, protein modification, and neutrophil extracellular trap (NET) formation. Many reversible and irreversible PAD4 inhibitors have been reported recently. In this review, we summarize the structure–activity relationships of newly investigated PAD4 inhibitors to bring researchers up to speed by guiding and describing new scaffolds as optimization and development leads for new effective, safe, and selective cancer treatments. In addition, some recent reports have shown evidence that PAD4 inhibitors are expected to trigger antitumor immune responses, regulate immune cells and related immune factors, enhance the effects of immune checkpoint inhibitors, and enhance their antitumor efficacy. Therefore, PAD4 inhibitors may potentially change tumor immunotherapy and provide an excellent direction for the development and clinical application of immunotherapy strategies for related diseases.

## 1. Introduction

Protein arginine deiminase (PAD, including PAD1-4 and PAD6) is one of the important post-translational modification (PTM) enzymes that convert protein arginine residues into non-coding citrulline residues in a calcium-dependent manner [1]. This review is focused on PAD4, the only isoform of PAD enzymes carrying a normalized nuclear localization sequence (NLS) [2], which is highly expressed in neutrophils and regulates the citrullination of a variety of proteins, such as histone H3 [3,4]. Citrullinated histone H3 (H3Cit) is associated with the formation of neutrophil extracellular traps (NETs), which are web-like chromatin structures complexed with intracellular neutrophil proteins, in a process known as NETosis [5,6]. Dysregulated citrullination and resulting excess NETosis are implicated in cancer progression and have been well investigated recently [7,8]. Increasing evidence suggests that PAD4 plays a pathophysiologic role in various diseases, making PAD4 an attractive therapeutic target.

Harnessing the immune system against cancer has been recognized as an integral protocol of standard oncology clinical practice and management. Thus, immunogenicity and immunotherapy have produced remarkable responses to several types of cancers [9,10]. However, despite impressive clinical results, many patients either develop resistance or relapse after treatment. This allows scientists to develop combinatorial regimens that precisely target barriers to effective antitumor immune responses, thereby improving therapeutic outcomes.

Recent studies have shown that epigenetic dysregulation hinders the effective immune responses in cancer therapies and leads to cancer resistance to immunogenicity and immunotherapy [11,12]. Epigenetic remodeling has been considered a major mechanism regulating cancer development and progression [13,14] and antitumor immunity [15]. Histone post-translational modification (PTM) is a major hallmark of epigenetic regulation [16]. Genetic lockout or chemical inhibition of PAD4 results in the inability of neutrophils to form citrullinate histones and failure in NETosis [17]. Meanwhile, abnormal increases in NETs promote immune rejection and inhibit T-cell-mediated antitumor immune responses [18]. Therefore, PAD4 is one of the indispensable members of the mammalian immune system. This review summarizes the application of PAD4 inhibitors in antitumor immunotherapy in recent years.

## 2. The Structure and Function of PAD4

The functional schematic of PADs catalyzing the conversion of peptidyl arginine residues to citrulline with the participation of calcium ions is shown in Figure 1A. The human PADI4 gene encodes a 74 kDa PAD4 monomer (663 amino acids) containing two major domains, namely an N-terminal structural domain (M1-P300) and a C-terminal catalytic domain (N301-P663) [19,20]. As shown in Figure 1B, the terminal end of the N-terminal structural domain includes a nuclear localization signal (NLS) motif (P56PAKKKKKST63) that promotes the passage of PAD4 through the nuclear pore [8]. The enzyme active site of PAD4 is located in the C-terminal domain and exhibits an α/β propeller structure [21]. Physiologically, PAD4 presents a head-to-tail homodimer conformation with full catalytic activity and Ca^2+^ binding [22]. Dimerization of PAD4 is stabilized by multiple hydrophobic interactions and intermolecular salt bridges (e.g., R8/D547), and the disruption of PAD4 dimer results in a loss of more than half of the enzymatic activity [23]. Human PAD4 includes five Ca^2+^-binding sites that structurally influence the active conformations of the PAD4 enzyme [19,20,24]. Ca^2+^(1) and Ca^2+^(2) are located in the C-terminal domain, while Ca^2+^(3), Ca^2+^(4), and Ca^2+^(5) are anchored in the N-terminal domain. PAD4 binding with calcium could allow a more than 10,000-fold increase in enzyme activity [23]. In the active cavity of PAD4, a negatively charged U-shaped channel is the binding site of the substrate (arginine) or corresponding inhibitors [24,25]. As shown in Figure 1C, in the binding site, C645, D350, D473, and H471 are essential residues for the conversion of arginine substrates to citrulline. D350 and D473 act as “anchors” that strongly interact with the guanidine moiety of the arginine substrate via two salt bridges. Then, a nucleophile (C645) attacks the guanidine carbon of arginine to form a covalent tetrahedral structure, which is stabilized by protonation of the tetrahedral intermediate by H471. Subsequently, the intermediate disintegrates to yield a covalent S-alkylthiourea intermediate and an ammonia molecule. S-alkylthiourea is finally hydrolyzed with a water molecule to yield citrulline and the original C645 thiol salt [20]. Notably, changes in any essential residue result in a significant decrease in enzyme activity, suggesting a highly synergistic process in citrullination with PAD4 [25].

PAD4 is widely expressed in human tissues, and in addition to hematopoietic stem cells and immune cells, PAD4 enzymes have been detected in the human brain, pituitary gland, uterus, joints, and bone marrow [26,27,28]. Tumors of different tissue origins, including hepatocellular carcinoma, lung cancer, breast cancer, colon cancer, and leukemia, have been found to overexpress PAD4 [8,29,30,31]. Intriguingly, PAD4 has the capacity to specifically target several nuclear proteins, such as ING4 (growth inhibitory factor 4) and histone H3 and H4 [8,32]. PAD4 is involved in the regulation of multiple physiological pathways, such as NETosis, gene regulation, and apoptosis [33], and it is especially involved in the regulation of aberrant citrullination leading to dysregulation of NETosis, which leads to disease progression or exacerbation. NETs are involved in the pathogenesis and progression of a variety of diseases, as shown in Figure 2.

## 3. Reversible (Non-Covalent) PAD4 Inhibitors

The great degree of structural conservation of the PAD active site across all isoforms complicates the design of inhibitors specific to PAD4 isoforms. Simultaneous inhibition of multiple PAD isoforms is undesirable because each isoform is involved in a different biological pathway [28]. Reversible PAD4 inhibitors have a wider range of chemical compositions and rely on various modes of PAD4 inhibition, including interactions with residues in the PAD4 active site and “front gate” and “back gate” occupations in the U-tunnel [39]. The structures and activities of reversible (Non-Covalent) PAD4 inhibitors are listed as shown in Figure 3 and Table 1. Early research concentrated on identifying a number of reversible PAD inhibitors, such as paclitaxel (Ki = 4.5–10 mM) and benzoyl-Nω, Nω-dimethylarginine (Bz-ADMA, IC_50_ = 0.4 mM), and the high-throughput screening tool for activity-based protein profiling (ABPP-HTS) screened for minocycline (Ki = ~0.78 mM), tetracycline (Ki = ~0.62 mM), chlortetracycline (Ki = ~0.11 mM), and sanguinomycin (Ki PAD4 = 80 µM) [21,40,41,42,43]. However, their utilization as PAD inhibitors is not promising because of their low selectivity and inhibitory activity only at high micromolar to millimolar concentrations. Among them, ruthenium red (**7**, Ki PAD4 = 10 µM) is also an effective inhibitor of other PAD isozymes [21].

Another study by Supuran’s group [44] found that the guanidine derivative **8** was demonstrated to reduce PAD4 activity (36% inhibition at 10 μM), comparable to the 35% inhibition of the control compound Cl-amidine (described later in the Irreversible Inhibitors section) [44]. Additionally, Ferretti et al. reported a novel PAD inhibitor **9** that contains a 3,5-dihydroimidazol-4-one ring to replace the acyclic guanidinium portion of the arginine residue [45]. This novel small-molecule PAD3 inhibitor showed high inhibitor activity at 100 nM, but activity against other PADs was not shown.

Lewis et al. reported a strong, reversible inhibitor with notable PAD4 selectivity [46]. In this work, the authors searched for PAD4 inhibitors in the presence and absence of calcium using a library of small molecules encoding DNA. Following lead compound optimization, GSK199 (**10**) and GSK484 (**11**) were discovered. Interestingly, their inhibition was competitive with calcium, and the compounds preferentially bound to calcium-free PAD4. With IC_50_ values of 250 nM and 50 nM, respectively, they inhibited PAD4 in the presence of 0.2 mM Ca^2+^, but their potency was more than five times lower in the presence of higher Ca^2+^ concentrations. However, comprehensive kinetic evaluations revealed a mixed mode of inhibition for these substances and revealed that, in contrast to other PADs, they were over 35 times more selective for PAD4 [41]. Following these studies, Gajendran’s team [47] screened JBI-589 (**12**) and found dose-dependent inhibition of PAD4 enzyme activity with an IC_50_ of 0.122 µM when tested against recombinant human PAD4 enzyme in an ammonia release assay at 10 semi-log concentrations. JBI-589 was tested in a comparable assay format against other human PAD enzymes and, even at the highest concentration tested (30 µM), no inhibitory activity was observed against these enzymes. This unequivocally demonstrates how highly selective JBI-589 is for PAD4. In an ELISA assay, JBI-589 treatment dose-dependently inhibited histone H3 guanosine chemotaxis induced by 25 µM calcium ion carrier in human neutrophils. JBI-589 displayed an EC_50_ of 0.146 µM in this assay [47]. In addition, JBI-589 is orally bioavailable in mice and possesses good ADME properties. The pharmacokinetic studies conducted on mice using intravenous and oral administration revealed half-lives of 8.0 and 6.3 h, respectively [48]. All things considered, these inhibitors are an excellent illustration of how well high-throughput screening can be combined with in-depth biochemical and structural characterization to produce novel compounds that may have therapeutic uses.

In some other studies, Tejo et al. discovered multiple new reversible PAD4 inhibitors, including peptidyl inhibitors containing furin (the most promising of which is Inh-Dap, **13**, with an IC_50_ of 243.2 ± 2.4 µM), through structure-based virtual screening [49]. Ardita and his colleagues identified six new compounds by computerized high-throughput screening methods [50]. The performance of these four potential PAD4 inhibitors is better than that of Cl-amidine (SC97362, **14**, was the most promising with an IC_50_ = 1.88 ± 0.26 µM) [50], and these four potential PAD4 inhibitors are expected to serve as design templates for reversible PAD4 inhibitors.

**Table 1 pharmaceutics-16-00335-t001:** IC_50_ values and in vivo activity for reversible PAD4 inhibitors.

Compound	Name	PAD4 Inhibition	In Vivo Activity (PAD4)	Ref.
**1**	Paclitaxel ^a^	Ki= 4.5–10 mM	-	[21,42]
**2**	Bz-ADMA ^a^	IC_50_ = 0.4 mM	-	[40,51]
**3**	Minocycline ^b^	IC_50_ = 0.62 ± 0.01 mM	-	[21,40,41]
**4**	Tetracycline ^b^	IC_50_ = 0.78 ± 0.14 mM	-	[43]
**5**	Chlortetracycline ^b^	IC_50_ = 0.10 ± 0.01 mM	-	[40,43]
**6**	Sanguinomycin ^b^	Ki PAD1 = 2000 µM PAD2 = 100 µM PAD3 = 60 µM PAD4 = 80 µM	-	[40]
**7**	Ruthenium red ^b^	Ki PAD1 = 30 µM PAD2 = 17 µM PAD3 = 25 µM PAD4 = 10 µM	-	[40,41]
**8**	8 ^a^	36% inhibition at 10 μM	-	[44]
**9**	9	IC_50_ (PAD3) = 100 nM	-	[45]
**10**	GSK199 ^a^	IC_50_ (0 mM Ca^2+^) = 200 nM IC_50_ (2 mM Ca^2+^) = 1.0 μM	Collagen-induced arthritis mouse model	[40,52,53]
**11**	GSK484 ^a^	IC_50_ (0 mM Ca^2+^) = 50 nM IC_50_ (2 mM Ca^2+^) = 250 nM	MMTV-PyMT mouse model for mammary carcinoma (FVB/n background); RIP1-Tag2 mouse model for pancreatic neuroendocrine carcinoma (C57BL/6 background); xenograft MDA-MB-231 mouse model	[40,52,54,55]
**12**	JBI589 ^a^	IC_50_ PAD4 = 122 nM PAD1 > 30 µM PAD2 > 30 µM PAD3 > 30 µM	Collagen-induced arthritis (CIA) model in DBA/1 J mice; mouse LL2, B16F10, and EL4 tumor models	[47,48]
**13**	Inh-Dap ^a^	IC_50_ = 243.2 ± 2.4 μM	-	[41,49]
**14**	SC97362 ^c^	IC_50_ = 1.88 ± 0.26 μM	-	[41,50]

^a^ IC_50_ values were determined by adding varying concentrations of benzoylarginine ethyl ester (BAEE) substrate to initiate the enzyme, pre-warming PAD4 and inhibitor in the presence of varying concentrations of calcium prior to the assay, stopping the reaction, and quantifying the amount of Cit produced, or quantifying the production of ammonia. ^b^ The PAD4-targeted activating protein profiling (ABPP) reagent RFA (rhodamine-conjugated F-amidine) has a fluorescent moiety. The test compounds compete with RFA for the binding of PAD4, and the PAD4 inhibitory activity was detected by measuring fluorescence. ^c^ Evaluation was performed using dansyl-Gly-Arg as substrate. The fluorescence emitted by dansyl was monitored, and the citrullination activity of PAD4 was calculated based on the peak area of the product of citrullinated dansyl-glycine-arginine.

## 4. Irreversible (Covalent) PAD4 Inhibitors

The mechanism of irreversible PAD inhibitors is specialized and involves covalent binding of cysteine residue (C645) in the catalytic active site. These inhibitors usually act on calcium-bound PAD. The structures and activities of Irreversible (Covalent) PAD4 inhibitors are listed as shown in Figure 4 and Table 2. On the basis of ABPP-HTS, Thompson et al. also identified NSC95397 (**15**) and streptavidin (**16**) as irreversible PAD inhibitors [56,57]. Significantly, streptavidin demonstrated remarkable potency and selectivity against PAD4, potentially because of its benzene ring and substituted pyridine group [21]. These inhibitors modify the active site cysteine covalently through the presence of α,β-unsaturated carbonyl functional groups.

Furthermore, the irreversible inhibition of PAD4 by 2-chloroacetamidine (**17**) was also confirmed by Thompson et al. [58,59], who invented Cl-amidine (**19**) and F-amidine (**20**), the first generation of irreversible pan-PAD inhibitors, by replacing the original guanidinium group with a haloacetamidine group, on the scaffold of the small-molecule substrate of PAD4, benzoyl-L-arginine amide (BAA, **18**) [60,61]. According to kinetic studies, both substances covalently alter the cysteine in the active site in a concentration- and time-dependent manner, thereby irreversibly rendering PAD4 and other PAD isoenzymes inactive when they are bound to Ca^2+^ [60,62]. Cl-amidine is more potent than F-amidine, which may be related to the fact that chlorine is a better leaving group than fluorine. Subsequently, Cl-amidine emerged as the most often utilized substance and served as a standard tool compound by which other cutting-edge PAD inhibitors were evaluated for effectiveness. It successfully prevented the formation of NETs and histone citrullination [63], and it promoted disease severity in several animal models [46,64]. In this way, the design of most of the new compounds is based on the Cl-amidine scaffold. Furthermore, by incorporating a carboxylic acid in the ortho-position of the phenyl group, the second-generation PAD inhibitors, o-Cl-amidine (**21**) and o-F-amidine (**22**), were developed. These compounds significantly increased the inhibitory effects and selectivity among PADs, and they improved the inhibition of PADs and H3 citrullination in HL-60 cells 100-fold over that of Cl-amidine [65].

In order to evaluate how the length of the chain connecting the haloacetamidine slug and peptide backbone affected the inhibition of PAD4, Thompson et al. synthesized X4-amidines and X2-amidines (X = Cl, F; **23–26**) with four- and two-methylene bridges, respectively [59,60,65]. These compounds were poor PAD4 inhibitors, probably because of the inability to properly localize the slugs. Therefore, subsequent designs of PAD4 inhibitors based on the Cl-amidine scaffold preferred three methylene bridges. To assess the impact of chirality, D-Cl-amidine (**27**), D-F-amidine (**28**), D-o-Cl-amidine (**29**), and D-o-F-amidine (**30**) were synthesized. Although these compounds were not as potent as the parent compounds, they were selective PAD1 inhibitors, suggesting that the chiral center can turn over isoform selectivity [66]. Additionally, they displayed improved pharmacokinetics, maximum tolerated dose, and bioavailability, which could be brought on by a decrease in protein hydrolysis and the production of harmful metabolites. A series of tetrazole-substituted analogs were synthesized based on the C-terminal bioisosteric replacement of Cl-amidine [67], among which some analogs exhibited increased potency and selectivity. Ortho-position carboxylic acid modification (**31**) resulted in a 30-fold increase in potency, underscoring the significance of this pharmacophore, a finding similar to the SAR results produced by the modification of carboxylic acids in o-Cl-amidine and o-F-amidine. It was also found that a butyl group (**32**) on the tetrazole ring enhanced cellular activity, whereas the modification of o-carboxylate showed low levels of cellular activity, which could result from the negatively charged carboxylate impeding the uptake by cells, and it is hypothesized that raising the hydrophobicity of the compounds can lead to improved cell permeability [67].

Alternatively, BB-Cl-amidine (**33**) and BB-F-amidine (**34**) were synthesized based on backbones of Cl-amidine and F-amidine, respectively, by merging a C-terminal amide with a benzimidazole group while an N-terminal amide was substituted with a biphenyl group [68,69]. BB-Cl-amidine (CLogP = 4.17) has significantly increased lipophilicity compared to Cl-amidine (CLogP = −0.23) and is predicted to facilitate cell entry [69]. In line with this predicted inhibition, these inhibitors exhibited similar potency to Cl-amidine and F-amidine in vitro. However, BB-Cl-amidine was approximately 20 times more cytotoxic than Cl-amidine against U2OS cells, a PAD4-expressing cell line (EC_50_ = 200 µM for Cl-amidine versus EC_50_ = 8.8 µM for BB-Cl-amidine). In addition, the in vivo half-life of BB-Cl-amidine was significantly longer than that of Cl-amidine, despite similar microsomal stability [68]. Not coincidentally, Wang and colleagues [70] synthesized a series of compounds, including YW356 (**35**), as potentiated pan-PAD inhibitors by modifying the Cl-amidine scaffold with Cα-amidino-toluene and Nα-amidino-dimethylnaphthylamine. Improved in vitro inhibition of PAD4 enzyme activity (IC_50_ = 1–5 µM) and markedly increased cytotoxicity against U2OS cells (IC_50_ of about 2.5 µM, 60 times greater than Cl-amidine) were demonstrated by YW356 [70]. Further mechanistic studies confirmed that YW356 inhibited cancer cell proliferation by inhibiting H3 citrullination and activating the expression of p53 target genes (including SESN2). Furthermore, YW356 has been demonstrated in a number of preclinical tumor models to be a strong inhibitor of cancer progress and metastasis [71,72,73]. According to recent studies, cation-penetrating peptide-modified gold nanoparticles can dramatically boost YW356’s cellular uptake and consequently boost its antitumor activity. This suggests the application of nanoparticles in the design of PAD4 inhibitors. Recently, Zhu et al. have developed an improved PAD4 inhibitor (ZD-E-1M, **36**), which is derived from YW356 by replacing Nα-amido-dimethylnaphthylamine with nitrobenzofuran. ZD-E-1M selectively inhibited the PAD4 enzyme in vitro with an IC_50_ of 2.39 µM and demonstrated strong antitumor and anti-metastatic effects (effective dose: 5 μmol/kg) in a mouse 4T1 breast cancer model [74]. Interestingly, ZD-E-1M is pH-responsive and can be self-assembled into nanoparticles that exhibit flower-like nanostructures in acidic solutions, which are more loosely packed than those at pH 7.4. The long-term accumulation of nanostructure at tumor sites is facilitated by flower nanostructures, which also deliver drugs selectively into tumor cells, increasing cytotoxicity. The introduction of nitrobenzofuran, on the other hand, also brings fluorescent properties to the PAD4 inhibitor, which is expected to be a tool for monitoring PAD4 activity in cells and tissues given the pathology associated with dysregulation of PAD4 activity.

Subsequently, Zhu et al. [75] further developed a range of highly targeted PBA-PAD4 inhibitors against tumor cells, using phenylboronic acid to modify the PAD4 inhibitors so that they could be specifically taken up by tumor cells. PBA-modified PAD4 inhibitors demonstrated a significant decrease in the formation of NETs in the tumor tissues and were able to inhibit the growth and metastasis of breast cancer in the in vivo 4T1-homozygous mouse model in a concentration-dependent manner. Compound 5i (**37**) was shown to have optimal antitumor activity [75]. The PBA modification ensured that 5i had in vivo safety with no significant damage to organs such as the liver and kidney, which offers a fresh perspective on developing highly targeted PAD4 inhibitors [75,76]. After that, our group [77] further constructed smart oxidative stress-responsive nanomedicine (K-CRGDV-4B, **43**) by covalently attaching the PBA-PAD4 inhibitor 4B (**42**) to RGD peptide-modified chitosan by utilizing the oxidative stress responsiveness of the phenylboronic acid moiety. The K-CRGDV-4B NPs were verified to increase drug accumulation of 4B at the tumor site and enable its responsive release in the tumor microenvironment (acidic conditions as well as excess H_2_O_2_). In the mouse Lewis lung cancer metastasis assay, tumor inhibition after treatment with 0.2 μmol/kg of K-CRGDV-4B was comparable to that of 5 μmol/kg of 4B, with a 25-fold difference in dose. These results suggest that K-CRGDV-4B NPs improve the drawbacks of fast metabolism and poor stability of PAD4 inhibitors and enhance their therapeutic effects. This intelligent responsive nano-drug delivery system with good biosafety also provides an idea for other defective chemotherapy drugs [63].

Apart from developing PAD inhibitors in the small-molecule domain, Thompson and colleagues also detected TDFA (**38**) in a collection of 264 metapeptides that had C-terminal ornithine conjugated with F- or Cl-acetamidine [78]. TDFA is a selective inhibitor of PAD4 (15-, 52-, and 65-fold more potent than PAD 1, 2, and 3, respectively), while the Cl-acetamidine analog (TDCA, **39**) is equipotent for PAD1 and PAD4. Notably, in HL-60 cells, TDFA demonstrated good potency in inhibiting histone H3 citrullination. At 1 nM, it was equivalent to 100 µM Cl-amidine, and at 100 nM, total inhibition was observed [78]. In another interesting study, Thompson et al. also utilized azobenzene’s Z/E isomer to adjust the inhibitor’s light-induced potency. When compound **40T** was exposed to 350 nm light, **40C** was formed, which increased the potency of the compound by almost ten times, whereas the activity of compound **41T** was reduced forty-five times [79]. This photopharmaceutical molecule could be activated by light in desired tissues to treat diseases in which PAD activity is dysregulated.

**Table 2 pharmaceutics-16-00335-t002:** IC_50_ values and in vivo activity for irreversible PAD4 inhibitors.

Compound	Name	PAD4 Inhibition	In Vivo Activity (PAD4)	Ref.
**15**	NSC95397 ^b^	k_inact_/K_I_ (M^−1^min^−1^) PAD1: 175 PAD2: 1600 PAD3: 9150 PAD4: 4530	-	[21,56,57]
**16**	Streptonigrin ^b^	IC_50_ = 2.5 ± 0.4 μM k_inact_/K_I_ (M^−1^min^−1^) PAD1: 3700 PAD2: 12,000 PAD3: 3500 PAD4: 440,000	-	[21,56,57]
**17**	2-chloroacetamidine	-	-	[58,59]
**18**	BAA	IC_50_ = 0.25 ± 0.06 mM	-	[60,62]
**19**	Cl-amidine ^a^	IC_50_ = 5.9 ± 0.3 μM k_inact_/K_I_ (M^−1^min^−1^) PAD1: 37,000 PAD2: 1200 PAD3: 2000 PAD4: 13,000	Collagen-induced arthritis mouse model; DSS mouse model of colitis; murine sepsis model; mouse model of periodontitis	[46,60,62,80,81]
**20**	F-amidine ^a^	IC_50_ = 22 ± 2.10 μM k_inact_/K_I_ (M^−1^min^−1^) PAD1: 2800 PAD2: 380 PAD3: 170 PAD4: 3000	-	[60,62]
**21**	*o*-Cl-amidine ^a^	IC_50_ = 2.2 ± 0.31 μM k_inact_/K_I_ (M^−1^min^−1^) PAD1: 106,400 PAD2: 14,100 PAD3: 10,345 PAD4: 38,000	-	[65]
**22**	*o*-F-amidine ^a^	IC_50_ = 1.9 ± 0.21 μM k_inact_/K_I_ (M^−1^min^−1^) PAD1: 180,900 PAD2: 7500 PAD3: 6700 PAD4: 32,500	-	[65]
**23–26**	X-n-amidine (X = F or Cl, n = 2 or 4) ^a^	IC_50_ > 520 µM	-	[59,60,65]
**27**	D-Cl-amidine ^d^	k_inact_/K_I_ (M^−1^min^−1^) PAD4: 1400	-	[66]
**28**	D-F-amidine ^d^	k_inact_/K_I_ (M^−1^min^−1^) PAD4: 130	-	[66]
**29**	D-o-Cl-amidine ^d^	k_inact_/K_I_ (M^−1^min^−1^) PAD4: 30	-	[66]
**30**	D-o-F-amidine ^d^	k_inact_/K_I_ (M^−1^min^−1^) PAD4: 50	-	[66]
**31**	R1 = R3 = H, R2 = COOH, X = Cl ^a^	poor cellular activity	-	[67]
**32**	R1 = tBu, R2 = H, R3 = Ph, X = Cl ^a^	MTT assay in U2OS cells, EC_50_ = 10 ± 2.5 μM	-	[67]
**33**	BB-Cl-amidine ^a^	k_inact_/K_I_ (M^−1^min^−1^) PAD1: 16,100 PAD2: 4100 PAD3: 6800 PAD4: 13,300	Lupus-prone MRL/lpr mice; canine and feline mammary cancer xenograft mice	[68,69,82]
**34**	BB-F-amidine ^a^	k_inact_/K_I_ (M^−1^min^−1^) PAD1: 900 PAD2: 1200 PAD3: 3400 PAD4: 3750	-	[68,69]
**35**	YW356 ^a^	IC_50_ = 1–5 μM	Mouse sarcoma S-180 xenograft model; mouse nasopharyngeal carcinoma model; A549 xenograft mouse model	[70,71,72,73]
**36**	ZD-E-1M ^a^	IC_50_ = 2.39 μM	Mouse S180 sarcoma model; orthotopic 4T1 breast cancer model; Lewis mouse model of lung cancer metastasis	[74]
**37**	5i ^a^	IC_50_ = 1.9 ± 0.65 μM	Mouse S180 sarcoma model; orthotopic 4T1 breast cancer model	[75]
**38**	TDFA ^a^	IC_50_ = 2.3 μM k_inact_/K_I_ (M^−1^min^−1^) PAD1: 1700 PAD2: 500 PAD3: 400 PAD4: 26,000	-	[78]
**39**	TDCA ^a^	IC_50_ = 3.4 μM k_inact_/K_I_ (M^−1^min^−1^) PAD1: 21,000 PAD2: 300 PAD3: 920 PAD4: 24,000	-	[78]
**40**	40T ^a^ 40C ^a^	k_inact_/K_I_ (40T) = 600 M^−1^min^−1^ k_inact_/K_I_ (40C) = 5970 M^−1^min^−1^	-	[79]
**41**	41T ^a^ 41C ^a^	k_inact_/K_I_ (41T) = 4520 M^−1^min^−1^ k_inact_/K_I_ (41C) < 100 M^−1^min^−1^	-	[79]
**42**	4B ^a^	IC_50_ = 1.89 ± 0.33 μM	Mouse S180 sarcoma model; orthotopic 4T1 breast cancer model; Lewis mouse model of lung cancer metastasis	[75,77]
**43**	K-CRGDV-4B	-	Lewis mouse model of lung cancer metastasis	[77]

^a^ IC_50_ values were determined by adding varying concentrations of benzoylarginine ethyl ester (BAEE) substrate to initiate the enzyme, pre-warming PAD4 and inhibitor in the presence of varying concentrations of calcium prior to the assay, stopping the reaction, and quantifying the amount of Cit produced, or quantifying the production of ammonia. ^b^ The PAD4-targeted activating protein profiling (ABPP) reagent RFA (rhodamine-conjugated F-amidine) has a fluorescent moiety. The test compounds compete with RFA for the binding of PAD4, and the PAD4 inhibitory activity was detected by measuring fluorescence. ^d^ RCA (rhodamine-conjugated Cl-amidine).

Overall, a number of PAD inhibitors with differences in potency, isozyme selectivity, pharmacokinetics, and pharmacodynamics have been developed. Although the data from the various studies are not directly comparable with each other, it is evident from the data that among the reversible inhibitors, compound **9**, designed by Ferretti and coworkers, has PAD3 inhibitory activity at the nM level, and it can be verified whether the same can be true for PAD4 activity. The GSK series is highly selective and strongly inhibits PAD4 up to the nM level, and it has excellent pharmacokinetic properties. Among the irreversible inhibitors, TDFA is a highly selective inhibitor of PAD4. Chloramidine-based scaffold designs have had the most extensive research, and many have been shown to have good ex vivo antitumor activity or other disease-modifying effects, but the majority of compounds on the market today are pan-PAD inhibitors, which have comparable potency and can block any or all of the active PAD isozymes. Therefore, it is still crucial to develop and characterize isozyme-selective PAD inhibitors.

## 5. Delivery Systems for PAD4 Inhibitors

Compared to drugs administrated alone, nanodelivery systems can increase the drug concentration in the target area, thus increasing drug utilization and efficacy as well as reducing adverse drug reactions [83]. The rapid metabolism and lack of oral activity of PAD4 inhibitors are shortcomings, and researchers have paid attention to nanodelivery systems for PAD4 inhibitors. The cation-permeable peptide RKKRRQRRR (peptide TAT) and gold nanoparticles were used by Song et al. [84] to modify the PAD4 inhibitor YW356 (**35**) in order to create 356-TAT-AuNPs, which could enhance the penetration of PAD4 inhibitors in solid tumors. Compared to YW356 and 356-AuNPs, 356-TAT-AuNPs had greater anticancer potency. They were also more readily absorbed by the cells and enhanced antitumor activity through upregulating apoptosis, triggering autophagy, and blocking histone H3 citrullination [84]. Gold nanorods with RGD peptides and PAD4 inhibitors have been generated by Lu et al. [73] for combinational cancer treatment with photothermal therapy and chemotherapy. By using external laser irradiation to target the tumor precisely, gold nanoparticle photothermal treatment can decrease systemic tissue biotoxicity and increase therapeutic efficacy [85]. The anticancer activity was examined using the S180 sarcoma model, and tumor sections confirmed the permeability of the tumor tissue and the H3cit inhibition of YW356-loaded NPs. The anticancer activity of YW356-loaded NPs was consistent with a 10-fold increase in YW356 used alone, and the YW356-loaded NPs significantly increased antitumor activity when compared to NPs alone. The combination of photothermal therapy with chemotherapy for A549 tumor-bearing mice demonstrated a synergistic effect that increased the efficacy of the treatment [73]. The strategy of gold nanoparticles partially addressed the defects of PAD4 inhibitors in bioavailability and efficacy. However, gold is of concern in terms of biodegradability and biocompatibility, and the non-covalent drug-carrying mode leads to drug leakage and is expensive to build.

Our group [77] used the phenylboronic acid portion’s oxidative stress responsiveness to covalently attach the PBA-PAD4 inhibitor 4B (**42**) to RGD peptide-modified chitosan in order to further construct smart oxidative stress-responsive nanomedicines (K-CRGDV-4B, **43**). This process is detailed in the section on irreversible PAD4 inhibitors. Chitosan is a biocompatible carrier and inexpensive, and the covalent drug-carrying approach avoids leakage of PAD4 inhibitors. Increased tumor targeting of K-CRGDV-4B was shown by in vivo pharmacological stimulation in response to release; this resulted in its accumulation at the tumor site and the release of more of the PAD4 inhibitor by oxidative stimulation. This is an all-around intelligent, biosafe, and responsive nanodelivery technology that can be used not only for PAD4 inhibitors but also for other chemotherapeutic medications that have poor targeting or excessive toxicity [77]. Whereas the biosafety of chitosan itself makes the therapeutic effect of the nanomedicine limited by the activity of the PAD4 inhibitor itself, the development of compounds with stronger PAD4 enzyme inhibitory activity would be advantageous. Meanwhile, the functionality of the nanomedicine needs to be improved, for example, by the introduction of fluorescent moieties that are promising tools for monitoring PAD4 enzyme activity in cells and tissues.

A CREKA-modified ROS stimulus-responsive liposomal system was designed by Sun et al. [86] in a different study. It encapsulated Cl-amidine (**19**) in self-assembled liposomal nanocarriers (C-Lipo/CA) with microthrombus targeting, NETs, and inhibition functions for the cyclic guanosine monophosphate–adenosine monophosphate synthase–interferon gene-stimulating factor (cGAS-STING) pathway, allowing for the targeting of ischemic lesions and stimulus-responsive drug release [86]. This strategy deserves our attention for the application of this liposome in other PAD4 inhibitors, while its low drug loading rate (4.15 ± 0.21%) needs to be considered.

## 6. PAD4 Inhibitors Enhance Antitumor Immunotherapy

The control of gene expression takes place through epigenetic modifiers, and post-translational modification (PTM) of histones is one of the hallmarks of epigenetic regulation. Epigenetic modifications drive T-cell differentiation and function [87], thereby contributing to antitumor immune responses. Therefore, it is not surprising that epigenetic modifications are associated with cancer immunotherapy. Neutrophils produce extracellular traps (NETs), which were identified as a component of the innate immune system, with the ability to release antimicrobial agents and directly immobilize pathogens [88]. The interactions between NETs and infiltrating immune cells are also slowly being unraveled. Kaltenmeier and coworkers [89] found that tumor cells promoted the formation of PD-L1-embedded NETs by secreting granulocyte colony-stimulating factor (G-CSF) and interleukin-8 (IL-8). PD-L1-embedded NETs result in CD8-positive T-cell dysfunction, which was evidenced by upregulation of PD-1, Tim3, and lymphocyte-activated gene-3 (LAG3), as well as downregulation of IL-2, IFNγ, and TNFα [89]. In addition, Wang et al. [90] found that NETs were accompanied by regulatory T cells (Tregs) which promote the differentiation of CD4-positive T cells and NETosis [61]. On the other hand, NETs were found to promote the immunosuppression of Tregs, NK cells, and CD8-positive T cells [61] (Figure 5).

Given these strong associations, the synergistic role of PAD4 inhibitors with immunotherapy in antitumor responses was also highlighted. Zhu and colleagues found increased infiltrating CD4-positive and CD8a-positive T cells in tumors of the S180-loaded mouse model treated with the PAD4 inhibitor ZD-E-1M (**36**). After further analysis using single-cell flow mass spectrometry (CyTOF) of the tumors of 4T1 orthotopic tumor-bearing mice, it was observed that ZD-E-1M was involved in the regulation of the tumor immune microenvironment (TIME), increasing the number of dendritic cells (DCs) and CD4-positive T cells but decreasing the number of myeloid-derived suppressor cells (MDSCs) and the abundance of LAG3 on various immune cells [74]. No significant changes were observed in B cells, CD8-positive T cells, macrophages, and bone marrow mesenchymal stem cells (p-MDSCs). CXC chemokine receptor 3 (CXCR3) of CD4 T cells and CD8 T cells of CXCR4 showed significantly increased expression. In addition, ZD-E-1M was also a potential CTLA4 inhibitor, enhancing in vivo antitumor activity in combination with anti-PD1 Ab (αPD1) [74]. Subsequently, in 4T1 orthotopic tumor-bearing mice, Zhu and colleagues found that tumors treated with PAD4 inhibitor 5i (**37**) had a significant increase in normal neutrophils but a decrease in aging neutrophils (Naged CD194hi | CD62Llo) [75], which can promote breast cancer metastasis by mediating NETs [91]. Furthermore, the proportion of M1 macrophages increased, suggesting that 5i can also regulate the macrophage polarization ratio and activate the immune microenvironment [75]. No significant changes were found in DC cells, M2 macrophages, CD8-positive T cells, B cells, G-MDSCs, M-MDSCS, and CD4-positive T cells. The same conclusion was obtained in our recent study [77]. In the Lewis mouse model of lung cancer metastasis, tumors treated with 4B (**42**) and K-CRGDV-4B (**43**) showed a significant increase in the proportions of B lymphocytes, CD4-positive T cells, and CD8-positive T cells, and the changes in the proportions of the immune cells suggested that 4B and K-CRGDV-4B improved the tumor immune microenvironment. There were no discernible alterations found in DC cells, neutrophils, or M1 and M2 macrophages. Furthermore, PD-1 expression was equally reduced in all three immune cells, and the inhibition of PD-1 enhanced antitumor immunity. A combined application with a PD1 antibody (αPD1) also showed better antitumor and anti-metastasis effects [77] (Figure 5).

Deng and colleagues described a novel mechanism by which PAD4 in neutrophils promotes cancer progression. They found that neutrophil PAD4 regulates neutrophil transport, an effect mediated by the transcriptional regulation of CXCR2, and that PAD4 expression was positively correlated with CXCR2 expression in neutrophils. Treatment with the PAD4 inhibitor GSK484 (**11**) for 4 h did not affect neutrophil viability but significantly reduced CXCR2 receptor expression in neutrophils [48]. Pharmacological inhibition of PAD4 using the PAD4 isoform-selective small-molecule inhibitor JBI-589 (**12**) in LL2 tumor-bearing mice resulted in decreased CXCR2 expression and blocked neutrophil chemotaxis. PAD4 deletion or inhibition of PAD4 by JBI-589 (**12**) reduced primary tumor growth and lung metastasis in mouse tumor models, significantly enhancing the effects of the immune checkpoint inhibitors, anti-CTLA-4 and anti-PD-1 antibodies [48] (Figure 5).

Another major advantage of PAD4 inhibitors is that inhibition of the PAD4 pathway does not lead to immunosuppression. Knockout studies clearly showed that PAD4-deficient mice remained normal with no increase in infections and no signs of immunomodulation compared to PAD4 wild-type mice [92]. Enhancing immunogenicity and the efficacy of immunotherapy requires targeting immunosuppressive signals to alleviate some of the resistance of monoimmunotherapy. Tumor cells release factors related to the tumor microenvironment (TME), leading to tumor metastasis and immune evasion. Commonly, PD-L1 binds to PD-1, leading to evasion of immune responses and thereby promoting tumor progression. Many monoclonal antibodies with FDA approval have been used in clinical settings to inhibit the function of immunological checkpoints. But later in the course of treatment, a lot of cancers that respond to immune checkpoint inhibitors develop resistance. The tumor microenvironment’s high concentration of immunosuppressive cells, which can severely limit the infiltration and activity of cytotoxic lymphocytes (CTLs) and promote tumor growth, is one of the factors contributing to the resistance phenomenon. As mentioned above, PAD4 inhibitors are able to inhibit the expression of genes, such as PD1 and LAG3, which are potential immune checkpoint inhibitors, and their ability to inhibit tumor progression can be enhanced in combination with anti-CTLA-4 and anti-PD-1 antibodies. Meanwhile, PAD4 inhibitors can enhance the tumor site’s microenvironment, lower the percentage of immune-suppressive cells, and boost CTL infiltration and function, thus improving immunosuppressant resistance. These results demonstrate the important role of PAD4 inhibitors in improving antitumor immunotherapy. Unlike most current therapies that result in immunosuppression, PAD4 inhibitors are significantly better tolerated, and the combination of PAD4 inhibitors and immunotherapies has shown great potential to transform cancer treatment and prognosis.

## 7. Conclusions

In this review, we described the structure and function of the PAD4 enzyme and summarized the recently reported PAD4 inhibitors and their structure–activity relationships. The currently reported PAD4 inhibitors are still in the preclinical research phase, although multiple pan-PAD inhibitors exhibited potential in various animal cancer models. Notably, there is a strong requirement for the development of next-generation reversible and irreversible PAD inhibitors with enhanced potency, selectivity, and bioavailability, while minimizing off-target effects and side effects. In addition, we also summarized the evidence reported in recent years that PAD4 inhibitors play an important role in immunomodulation. However, the complexity of the tumor immune microenvironment and the role of PAD4 inhibitors in antitumor immunity of different immune cells have not been fully evaluated. Therefore, scientists need to conduct more studies to determine the role of PAD4 inhibitors in immunotherapy. We expect this review will enhance the understanding of the critical role of PAD4 in cancer progression and immunotherapy as well as provide an open perspective for the development of next-generation PAD4 inhibitors with dual functions of antitumor activity and antitumor immunity.

Take-Home Message: PAD4 inhibitors have antitumor activity and enhance antitumor immunity.

## Figures and Tables

**Figure 1 pharmaceutics-16-00335-f001:**
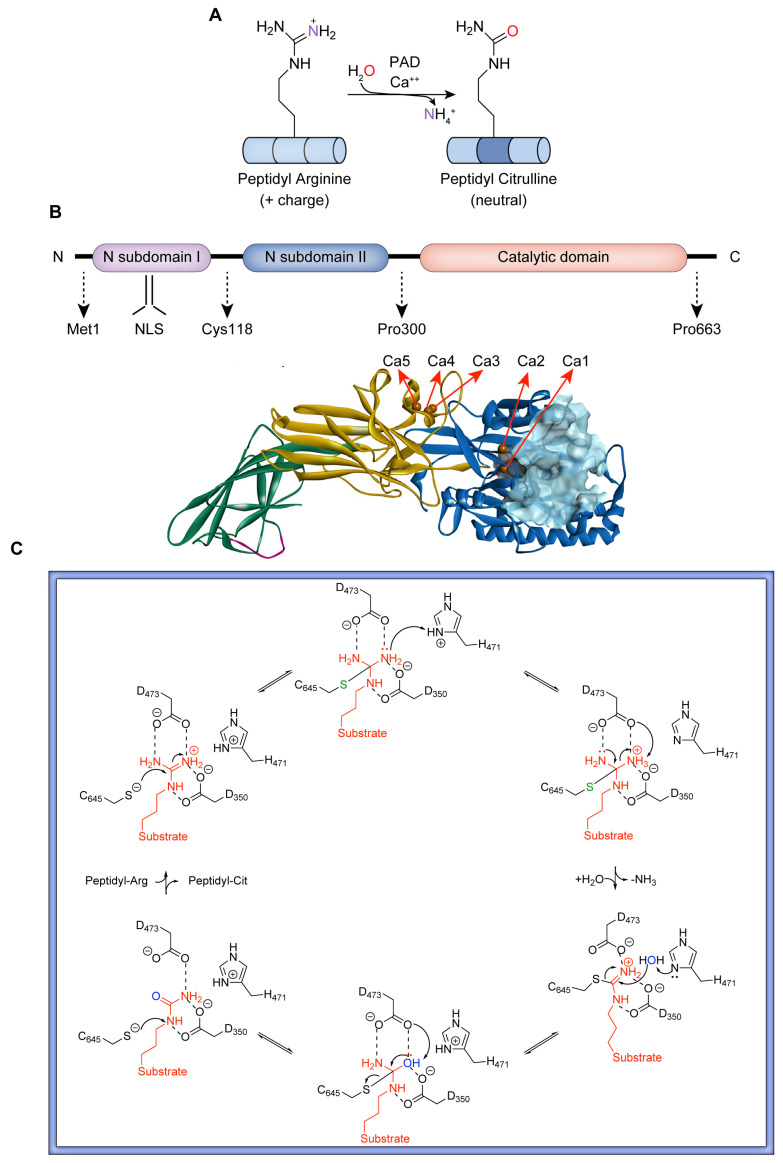
(**A**) PAD-mediated citrullination: conversion of arginine to citrulline in proteins catalyzed by PAD with the involvement of calcium ions. (**B**) Structure of the calcium-bound PAD4 monomer (PDB: 1WD9). (**C**) Proposed catalytic mechanism of PAD4.

**Figure 2 pharmaceutics-16-00335-f002:**
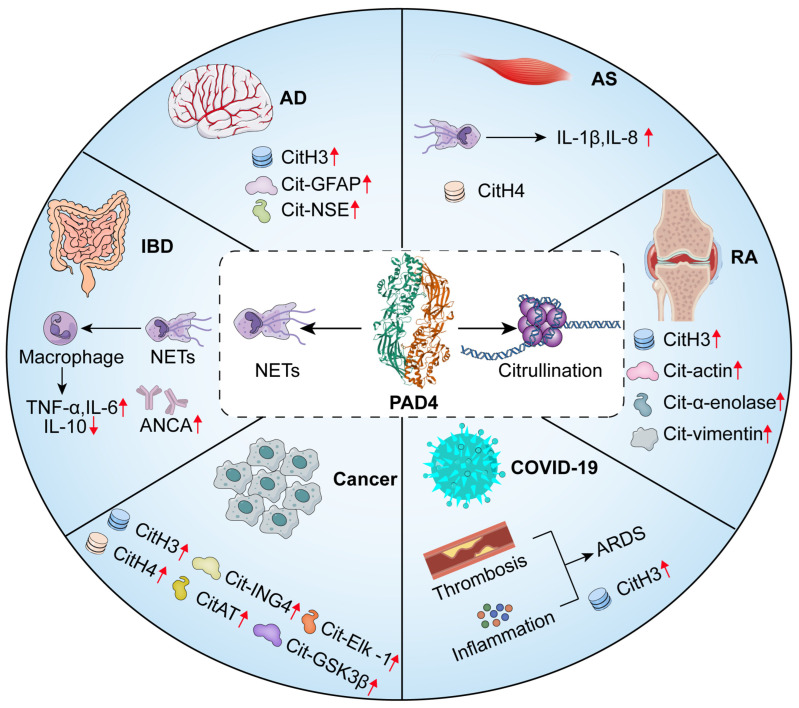
PAD4 induces citrullination and NETs in human disease. Alzheimer’s disease (AD): aberrant expression of CitH3, Cit-GFAP (Glial Fibrillary Acidic Protein), and Cit-NSE (neuron-specific enolase) [34]; inflammatory bowel disease (IBD): NETs induce an increase in pro-inflammatory cytokines, a decrease in anti-inflammatory factors, and an increase in ANCAs (anti-neutrophil cytoplasmic antibodies) [35]; COVID-19: higher levels of NETs trigger an inflammatory response and vascular microthrombosis, leading to ARDS (acute respiratory distress syndrome) [36]; rheumatoid arthritis (RA): aberrant citrullination of actin, histone H3, α-enolase, and waveform protein in RA [15,37]; atherosclerosis (AS): NETs can directly induce endothelial cell dysfunction through derived proteases, and activation of histone H4 leads to AS plaque destabilization [28,38]; cancer: CitH3, CitH4, ING4, CitAT (Antithrombin), GSK3β (Glycogen synthase kinase 3β), Elk -1 (Recombinant Human Guanylate Kinase), and others [8,32]. The red arrows represent abnormal expression of protein increased or decreased.

**Figure 3 pharmaceutics-16-00335-f003:**
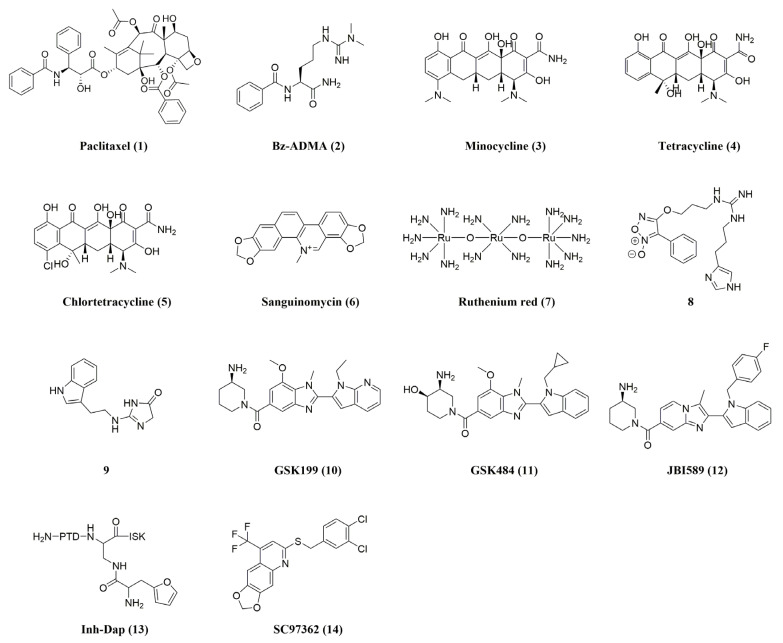
Structures of reversible PAD4 inhibitors.

**Figure 4 pharmaceutics-16-00335-f004:**
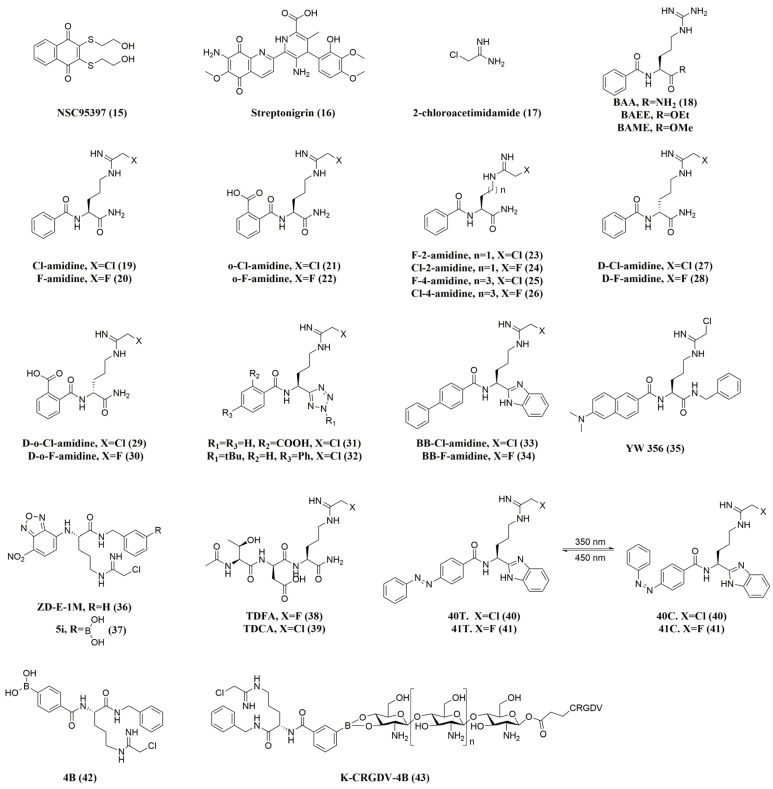
Structures of irreversible PAD4 inhibitors.

**Figure 5 pharmaceutics-16-00335-f005:**
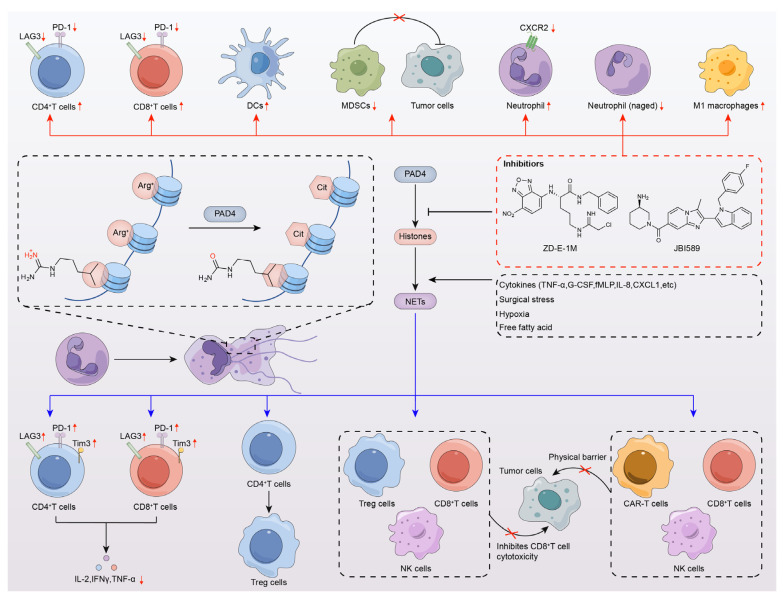
Novel role of PAD4 inhibitors in modulating antitumor immunity and immunotherapy. PAD4 inhibitors are directly involved in the regulation of the tumor immune microenvironment: (1) PAD4 inhibitors increase the expression of CD4^+^ T cells and CD8^+^ T cells and decrease the abundance of LAG3 and PD1 on their surface; (2) PAD4 inhibitors increase the expression of dendritic cells (DCs); (3) PAD4 inhibitors decrease the proportion of myeloid-derived suppressor cells (MDSCs) and indirectly lead to tumor cell suppression; (4) PAD4 inhibitors increase the expression of neutrophils, reduce their surface CXCR2, and reduced aged neutrophils (Naged); (5) PAD4 inhibitors increase the proportion of M1 macrophages; and so on. Indirect effects of PAD4 inhibitors on the immune microenvironment by inhibiting the production of NETs: (1) NETs lead to CD8^+^ T-cell and CD4^+^ T-cell dysfunction as evidenced by increased levels of PD-1, Tim3, and LAG3 as well as decreased production of IL-2, IFNγ, and TNFα; (2) NETs promote the differentiation of naive CD4^+^ T cells to Tregs; (3) NETs promote the immunosuppressive function of Tregs, NK cells, and CD8^+^ T cells; (4) NET-mediated physical barriers reduce the contact of CD8^+^ T cells, NK cells, and CAR-T cells, among others, with tumor cells, and so on.

## Data Availability

Not applicable.

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
