# Peer review of "Structure–Activity Relationship of PAD4 Inhibitors and Their Role in Tumor Immunotherapy"

_pharmaceutics, 2024, doi:10.3390/pharmaceutics16030335_

Round 1

Reviewer 1 Report

Comments and Suggestions for Authors

The review article by Jia et al. underscores the crucial role of PAD4 inhibitors in the era of Immunotherapy. As a key enzyme in post-translational modification, PAD4 controls various processes such as apoptosis, innate immunity, and pluripotency. Dysregulation of PAD4 has been linked to numerous diseases. While the review was well-written, there are some additions that could make it more informative and appropriate for the reader.

The first section of the review, which discusses inhibitors, was well-written. However, to enhance the reader's understanding, it is crucial to mention the Figure 1A in the text, and mark the PDB picture 1B to highlight all the domains.

Further, genome-wide association and pathology studies have revealed the critical role of PAD4 in the etiology of various diseases. Thus, a detailed paragraph about Expression/Localization and Transcriptional regulation is necessary; this should focus on its role in NET. A visual representation of how PAD4 citrullinates various substrates to mediate physiological functions would also be helpful.

It is also essential to elaborate on the role of PAD4 in Tumor immunotherapy, describing separately its role in different types of cancer. The author should also explain how its function is regulated with different ethnicities. If there are any inhibitors currently in clinical trial phase, these should be mentioned. If not, the shortcomings that need to be addressed should be outlined.

Finally, to make the review more catchy and memorable for the reader, a take-home message should be added after the conclusion.

Comments on the Quality of English Language

Minor editing is recommended.

Reviewer 2 Report

Comments and Suggestions for Authors

In this review, authors summarize knowledge on Protein Arginine Deiminase 4 (PAD4), its role in regulation of tumor immunology and on inhibitors under development.

Structure and function are molecularly well described. The list of inhibitors is exhaustive and their characterization well explained. However, remarks are:

As mentioned by authors on page 11 (line 267) all studies on inhibitors are not fully comparable. However, the authors could at least add a chapter on the assay(s) used for identifying compounds as PAD4 inhibitors. By explaining a little the different assays available, it would be easier for the reader to compare the activities listed in Tables 1 and 2.

Regarding inhibitors, only in vitro activities are presented in tables. Are some compounds more advanced ? If yes, another table for underlining the more advanced agents could be presented with specification of their stage (in vitro, in vivo, preclinical, phase 1, …). Many targets could be presented for treating cancer and many approaches fail. The fact to show advanced studies could reinforce the interest for such a review.

Some agents, like paclitaxel, are already used for treating cancer based on completely different mode of actions. Do the authors consider that their action on PAD4 is an additive mode of action ? Based on their IC50, is their activity on PAD4 anecdotical or to consider for using these drugs ?

Other PADs are mentioned, however, their functions are not described and just referenced in 26. However, the authors should detail a bit this part because knowledge on functions of the other PADs will help to estimate the risk of adverse events due to non or weak specificity of inhibitors.

Modulation of immune response by PAD4 is detailed and supported by a Figure but application(s) for treating tumors remains to be clarified based on all the information provided in this review:

What could be the more appropriate tumors to treat ? solid tumors are usually not infiltrated by neutrophils which remains around the tumors?

Do  the authors propose to used PAD4 inhibitors alone or in combination ?

Reviewer 3 Report

Comments and Suggestions for Authors

Dear Authors,

In your comprehensive review on PAD4 inhibitors, I would like to highlight two critical aspects that warrant further exploration and elucidation:

1. It is imperative to delve into the specific tumor types where PAD4 inhibitors exhibit their therapeutic potential. Authors should discuss the nuanced interplay between PAD4 inhibitors and the tumor microenvironment. How do these inhibitors modulate immune responses within the tumor context? Additionally, consider addressing whether environmental factors—such as hypoxia, acidity, or immune cell infiltrates—affect the specific anti-tumor responses elicited by PAD4 inhibitors. A detailed examination of this scenario will enhance our understanding of their clinical applicability.

2.  While PAD4 inhibitors hold promise, their effective delivery remains a critical challenge. Authors should emphasize the role of delivery systems in maintaining inhibitor activity. Consider exploring various delivery modalities, such as nanoparticles, liposomes, or targeted carriers. What are the advantages and limitations of each system? Highlighting the drawbacks of current delivery approaches will guide future research toward optimizing PAD4 inhibitor administration.

Round 2

Reviewer 1 Report

Comments and Suggestions for Authors

The authors have sufficiently improved the manuscript and revised optimally to address the comments made. Hence I recommend it for the publication.

Comments on the Quality of English Language

Please check for minor spelling correction.

Reviewer 3 Report

Comments and Suggestions for Authors

Authors have addressed the concerns raised in the review. The article can be accepted in the present form.